# Role of Akt Activation in PARP Inhibitor Resistance in Cancer

**DOI:** 10.3390/cancers12030532

**Published:** 2020-02-25

**Authors:** Ferenc Gallyas, Balazs Sumegi, Csaba Szabo

**Affiliations:** 1Department of Biochemistry and Medical Chemistry, University of Pecs Medical School, 7624 Pecs, Hungary; balazs.sumegi@aok.pte.hu (B.S.); szabocsaba@aol.com (C.S.); 2Szentagothai Research Centre, University of Pcs, 7624 Pecs, Hungary; 3HAS-UP Nuclear-Mitochondrial Interactions Research Group, 1245 Budapest, Hungary; 4Chair of Pharmacology, Department of Medicine, University of Fribourg, 1700 Fribourg, Switzerland

**Keywords:** PARP-Akt interplay, PI3K, mTOR, cytoprotection, apoptosis resistance, oxidative stress, mitochondrial protection

## Abstract

Poly(ADP-ribose) polymerase (PARP) inhibitors have recently been introduced in the therapy of several types of cancers not responding to conventional treatments. However, *de novo* and acquired PARP inhibitor resistance is a significant limiting factor in the clinical therapy, and the underlying mechanisms are not fully understood. Activity of the cytoprotective phosphatidylinositol-3 kinase (PI3K)-Akt pathway is often increased in human cancer that could result from mutation, expressional change, or amplification of upstream growth-related factor signaling elements or elements of the Akt pathway itself. However, PARP-inhibitor-induced activation of the cytoprotective PI3K-Akt pathway is overlooked, although it likely contributes to the development of PARP inhibitor resistance. Here, we briefly summarize the biological role of the PI3K-Akt pathway. Next, we overview the significance of the PARP-Akt interplay in shock, inflammation, cardiac and cerebral reperfusion, and cancer. We also discuss a recently discovered molecular mechanism that explains how PARP inhibition induces Akt activation and may account for apoptosis resistance and mitochondrial protection in oxidative stress and in cancer.

## 1. Introduction

Inhibitors of the constitutive mammalian enzyme poly(ADP-ribose) polymerase (PARP)1 received renewed interest in recent years due to their importance in cancer therapy [1]. Mostly, their use is based on the synthetic lethality theory [2], i.e., to block the single-strand DNA break repair process in double-strand DNA break repair (BRCA1/2) deficient cancer cells that selectively eliminates the cancer cells via DNA damage accumulation while not affecting normal cells possessing intact double-strand DNA break repair system [3,4]. Accordingly, review articles focusing on the role of PARP inhibitors in DNA repair vastly dominate over those that at least mention the effect of PARP inhibition on kinase signaling, although the latter effect is an important limiting factor in the cancer therapy [5], and a pivotal mechanism in non-cancerous therapeutic applications of the PARP inhibitors [6].

The phosphatidylinositol-3 kinase (PI3K)-Akt-mammalian target of rapamycin (mTOR) pathway is very often activated in human cancer [7]. The pathway is often utilized by various cytokine and growth factor receptor signaling, and mutation, expressional change or amplification of any of its elements could cause resistance against tumor therapy [7,8]. Accordingly, all members of this pathway (including Akt) were suggested as targets in monotherapy or combination therapy for various cancers. In turn, several inhibitors of the pathway were approved by the Food and Drug Administration [9]. In previous research papers, we demonstrated that PARP inhibition leads to Akt activation resulting in cyto- and mitochondria protecting actions [10]. In turn, we proposed that Akt inhibitors may be used to prevent limitation of PARP inhibitor based anti-cancer therapy [11]. Moreover, we have recently unveiled a novel molecular mechanism for the Akt activating effect of PARP inhibition [12]. In the present review, we seek to increase awareness of the therapeutic limitations that the PARP-Akt interplay represent. In addition, we discuss the role of the PARP-Akt interplay in non-oncological disease conditions and their experimental models.

## 2. The PARP-Akt Interplay

### 2.1. Cellular Role and Regulation of Akt

Protein kinase B/Akt, a 57-kDa serine/threonine kinase, is regarded as a pivotal mediator of cell survival promoting effects of various growth factors and other factors, including cAMP, hypoxia, and cytokines (Figure 1). In mammals, three Akt genes were identified that are differentially expressed at both the mRNA and protein levels [13]. Akt1 can phosphorylate thereby inactivate various components of the apoptotic machinery, therefore, it is involved in cellular survival pathways. Akt2’s function seems to be closely associated with the insulin receptor signaling pathway while the specific role of Akt3 is less clear [14]. Structure of Akts’ amino terminus pleckstrin homology (PH) domain and central kinase domain is conserved across evolution, while the carboxyl terminus hydrophobic and proline-rich domain is more variable. The kinase activity of Akt is regulated by phosphorylation of Thr^308^ and Ser^473^ situated in the PH domain, although phosphorylation of Ser^124^ and Thr^450^ sites are needed for sensitizing Akt toward regulatory stimuli [13]. In the classical activation pathway of Akt (Figure 1), the upstream element PI3K phosphorylates phosphatidylinositol-4,5-bisphosphate to phosphatidylinositol-3,4,5-trisphosphate (PIP3) that binds to PH domain of the constitutively active 3-phosphoinositide-dependent protein kinase (PDK)1. Binding of PIP3 to PDK1 results in translocation of the enzyme to the cell membrane’s inner surface, a crucially important step in Akt’s activation process. PIP3’s binding to Akt’s PH domain renders Akt more accessible to the PDK1 mediated phosphorylation of Thr^308^ regulation site [13]. Negative regulation of the pathway is mediated by the phosphatase and tensin homolog PTEN that removes the 3-phosphate from PIP3 [15].

Besides the classical pathway, Akt can be activated by various signaling processes. PDK1 can interact with the downstream element protein kinase C related kinase2, rendering it capable to phosphorylate both Thr^308^ and Ser^473^ of Akt1 in a PIP3 dependent manner [16]. The mammalian target of rapamycin complex 2 (mTORC2) that phosphorylates Ser^473^ is a PDK1 independent crucial mechanism of Akt activation (Figure 1). Additionally, mTORC2 and cyclin-dependent kinase 2/cyclin can phosphorylate Ser^477^ and Thr^479^, also resulting in Akt activation [17]. Integrin-linked kinase and calcium/calmodulin-dependent kinase-kinase that responds to the increases in intracellular calcium, activate Akt also by phosphorylating Ser^473^ [18,19]. When activated, Akt can phosphorylate various downstream targets that contain the consensus phosphorylation sequence RXRXXS/T, although Akt’s substrate specificity also depends on sequence determinants outside the consensus site.

By phosphorylating its downstream targets, Akt can regulate metabolism, apoptosis, transcription factors, angiogenesis, and cell cycle progression (Figure 1). The constitutively active glycogen synthase kinase (GSK)-3 phosphorylates thereby inactivates glycogen synthase, the enzyme responsible for the synthesis of glycogen from excess glucose in the liver. When high glucose levels in the blood triggers insulin secretion, the insulin signaling activates Akt that inactivate GSK-3 by Ser^9^ phosphorylation enabling glycogen synthesis that contribute to decreasing the blood glucose level [20]. In parallel, Akt activation via Akt-targeted Rab GTPase-activating proteins and downstream Rab GTPases along with the input of Rac1 and actin filaments, molecular motors, and membrane fusion regulators leads to cell surface expression of glucose transporters GLUT4 that allow the entry of excess blood glucose into muscle and adipose tissues (Figure 1). Attenuation of these processes results in symptoms of metabolic syndrome and type 2 diabetes [21].

Activated Akt regulates various downstream signaling proteins that promote cell survival [22]. Phosphorylation thereby inactivation of the pro-apoptotic B-cell lymphoma 2 (Bcl-2) family member Bcl-2 associated agonist of cell death (Bad) by Akt prevents Bad to form heterodimers with other pro-apoptotic Bcl-2 family members, translocate to the mitochondria, and, by labilizing its outer membrane, facilitate the release of cytochrome C, apoptosis inducing factor and endonuclease G from the mitochondrial intermembrane space (Figure 1). Instead, phosphorylated Bad forms a complex with the cytoplasmic scaffolding protein 14-3-3, shifting the balance toward dominance of anti-apoptotic Bcl-2 family members, stabilization of mitochondrial integrity, and promoting cell survival [23]. Besides blocking cytochrome C mediated activation of caspase 9, Akt effectively inactivates it by directly phosphorylating its Ser^196^ [24].

Inactivation of GSK-3 by Akt is another major survival pathway since, in addition to its metabolic role, GSK-3 extensively participates in signaling pathways. Under hypoxic condition, its Tyr^216^ is phosphorylated, resulting in activation of the enzyme [25]. The activated GSK-3 contributes to hypoxic-ischemic tissue injury by multiple mechanisms. It downregulates nuclear factor erythroid 2-related factor 2 (Nrf2) expression, Nrf2 translocation from the cytoplasm to the nucleus, and Nrf2 binding to antioxidant response element (ARE) DNA sequence, thereby limiting expression of antioxidant proteins encoded in Nrf2/ARE regulated genes (Figure 1) [26]. Apart from its pro-oxidative activity, GSK-3 activation causes decreased nuclear translocation of cAMP response element-binding protein (CREB) transcription factor [27], leading to increased expression of pro-inflammatory cytokines via altering the interaction of CREB and the pro-inflammatory transcription factor nuclear factor (NF)κB with the co-activator CREB binding protein (CBP) in the nucleus [28].

After activation, Akt translocates to the nucleus where it modulates by phosphorylation the activity of various transcription factors [29]. When phosphorylated by Akt, forkhead family transcription factors are retained in the cytoplasm in form of a complex with the 14-3-3 protein and cannot promote expression of genes encoding growth factors, stress response proteins, and enzymes of carbohydrate and lipid synthesis [30]. Akt not only prevents GSK-3’s negative effect on CREB activation [28], it phosphorylates CREB at its Ser^133^ that enhances CREB’s binding to CBP thereby increases CREB-mediated transcription of critical survival genes [31]. Inhibitor of NFκB (IκB) retains NFκB in the cytoplasm by forming a complex with it. Phosphorylation by its kinase (IKK), causes the release of NFκB from the complex, and marks IκB for ubiquitination and proteosomal degradation. Akt regulates the process by phosphorylating IKK at Thr^23^ thereby activating it that results eventually in expression of NF-κB dependent anti-apoptotic genes [32,33]. Additionally, Akt activates mTOR, a major node of the PI3K-Akt pathway, by direct phosphorylation. Activation of mTOR trigger translation initiation factors, hypoxia and angiogenesis associated factors, and cell survival and growth associated transcription factors [34]. All in all, the PI3K-Akt pathway promotes cell proliferation and survival by various mechanisms, therefore, hyperactivation of the pathway leads to apoptosis resistance and malignant transformation [13,35].

### 2.2. Nuclear Effects of PARP Activation

PARP1, the original (and principal) member of a family of 18 enzymes [36], binds to various DNA structures, including single- and double-strand breaks, crossovers, cruciforms, supercoils, and some specific double-stranded sequences [37]. This binding stimulates PARP1’s very low basal enzymatic activity. It cleaves its substrate, NAD^+^ to nicotinamide and an ADP-ribose unit, which it covalently attaches on glutamate, aspartate, arginine, lysine, or serine residues of target proteins. PARP1 can further catalyze formation of 2′-1″-O-glycosidic or α(1‴–2″)-ADP-ribose bonds, resulting in linear or branched ADP-ribose polymer (PAR) chains, respectively [38]. PAR chains are heterogeneous either in length (up to 200 ADP-ribose units) or in the extent of branching (one branch per 20–50 ADP-ribose units), however the role of this heterogeneity has not yet been elucidated [39]. PAR chains are catabolized very shortly after their synthesis to free ADP-ribose units by exo- and endoglycosidase activities of poly-(ADP-ribose) glycohydrolases (PARGs) [39]. The mono(ADP-ribose) unit bound directly to the protein that cannot be removed by PARG is cleaved off by terminal ADP-ribose protein glycohydrolases (TARGs) [40]. The PARP1 enzyme exists in a million-copy number in the nucleus and is responsible for the majority of PARylation occurring in mammalian cells [41]. In addition to the nuclear PARP, there is also a mitochondrial form of this enzyme in many cell types, the functional role of which is currently less understood [42]. The targets of PARP1’s enzymatic activity include core histones, the linker histone H1, a variety of transcription-related factors, and PARP1 itself, which is the primary target in vivo [43]. Due to its localization, activation characteristics, and enzymatic activity, PARP1 is considered to play a pivotal role in various DNA repair systems, and in the maintenance of genome integrity [44]. When PARP1 is knocked out, PARP2 takes over its role and vice versa, however the PARP1/PARP2 double knock out mutation results in embryonic lethality [41].

Recent reports indicate a scaffolding role for PARP1 in genotoxic stress induced apoptotic response [45,46]. PARP1 activated by DNA strand breaks PARylates itself and ataxia telangiectasia mutated kinase (ATM) to recruit a small ubiquitin-like modifier (SUMO)-1 ligase, PIASy, and nuclear IKKγ (NEMO). Concertedly, PIASy SUMOylates and ATM phosphorylates NEMO, leading to IKK activation and NFκB regulated resistance toward apoptosis [45].

Because of PARP1’s high sensitivity toward DNA breaks and its copiousness, genotoxic stress can induce hyperactivation of the enzyme leading to deleterious consequences for the organism [47]. Experimentally induced massive DNA damage triggers excessive PARP1 activation that causes depletion of the substrate NAD^+^ within minutes [47]. Therefore, the metabolic pathways that utilize NAD^+^ as substrate such as glycolysis, citric acid cycle and mitochondrial respiration become compromised, resulting in reduced ATP production and cellular dysfunction. To aggravate the precarious metabolic situation, phosphoribosyl pyrophosphate synthetase and nicotinamide mononucleotide adenylyl transferase consume ATP to resynthesize NAD^+^ leading to ATP depletion and predominantly necrotic cell death [48,49]. This mechanism (Figure 2) explains the detrimental effect of DNA-damaging agents, such as high dose of N-methyl-N’-nitro-N-nitrosoguanidine (MNNG) on cultured cells, and is consistent with the therapeutic efficacy of PARP-1 inhibition in animal models of oxidative stress related diseases. However, some studies indicated the necessity of including other mechanisms to explain the cytoprotective effect of PARP inhibition. Namely, MNNG was reported to cause similar NAD^+^ depletion in PARP1^−/−^ and PARP1^+/+^ fibroblasts [50], and PARP1 inhibition failed to prevent NAD^+^ and ATP depletion in carcinogen-treated hamster cells [51].

### 2.3. Role of the PARP-Akt Interplay in Shock and Inflammation

PARP is known to regulate a host of signal transduction pathways and PARP inhibitors have been shown to have the ability to suppress the production of various pro-inflammatory mediators (including cytokines and chemokines). These effects are thought to be important contributors to the beneficial effects of PARP inhibitors in various experimental models of local and systemic inflammation, as well as in various models of critical illness [52,53,54,55,56,57,58,59]. However, in addition to effects on pro-inflammatory signaling, PARP inhibitors’ anti-inflammatory and beneficial effects in various models of inflammation and shock may also be related, at least in part, to activation of Akt. In 2003 we reported that the protective effects of the PARP inhibitor PJ34 in a model of endotoxic shock in mice (which included improved survival and protection against multiorgan dysfunction) were associated with Akt activation in various organs including the liver and the gut [60], and similar results were subsequently obtained with another PARP inhibitor, 4-hydroxyquinazoline [61]. Moreover, in a model of neuroinflammation and demyelination in mice (induced by the mitochondrial toxin cuprizone), the PARP inhibitor 4-hydroxy-quinazoline protected against many of the observed pathophysiological changes (weight loss, demyelination): these effects were associated with a modulation of Akt activation [62]. In this model, the effect of cuprizone alone was associated with an increase in Akt phosphorylation; however, Akt phosphorylation was further and markedly enhanced in the presence of the PARP inhibitor [62]. It is conceivable that Akt activation contributes to the beneficial effect of PARP inhibition in these models; however, this remains to be directly tested in future experiments.

Additional inflammatory conditions where PARP-inhibitor/Akt “connection” has been evaluated include various in vitro and/or in vivo models of arthritis and dermatitis. In an experiment by Garcia and colleagues utilizing CD95/Apo-1 (Fas)-induced apoptosis in fibroblast-like synoviocytes (FLS) from rheumatoid arthritis patients, PARP-1-deficient FLS were protected from cell death compared to wild-type cells. This was associated with the activation of the Akt-GSK survival pathway in the PARP1-deficient cells, but inhibition of PI3K/Akt pathway did not affect the difference between PARP1-competent or -deficient cells in terms of viability [63]. Thus—although in this experimental setting the same authors have demonstrated that Akt activation can, in fact confer cytoprotection [64]—in these experimental conditions, pathways other than Akt must be responsible for the beneficial effect of PARP1 deficiency. In contrast, in models of sulfur mustard induced dermatitis, the beneficial effects of the PARP inhibitor ABT-88, as well as the beneficial effects of PARP1 deficiency, appears to involve a significant modulation of the Akt pathway. In these experiments, sulfur mustard exposure decreased Akt phosphorylation in the control group, while in the PARP1/deficiency + sulfur mustard groups, a marked increase in Akt phosphorylation was observed [65].

### 2.4. Role of the PARP-Akt Interplay in Reperfusion Injury

In one of the earliest studies related to the subject of PARP-Akt interplay, PARP inhibitors of two different structural class (HO-3089 and 4-hydroxyquinazoline) were evaluated on energy metabolism, contractile function, and cell death of isolated rat hearts during ischemia-reperfusion. In line with earlier studies demonstrating the beneficial effects of PARP inhibitors (or genetic PARP1 deficiency) in myocardial ischemia-reperfusion [66,67,68], both of the tested PARP inhibitors significantly improved the recovery of high-energy phosphate intermediates in the heart [69], directly confirming the well-known PARP-activation-cellular energetic failure concept. Both of the tested PARP inhibitors also improved cardiac contractility [69]. In addition, ischemia-reperfusion induced a slight, but detectable phosphorylation of Akt at Ser437 (i.e., evidence of Akt activation). Importantly, treatment with either of the two tested PARP inhibitors induced a marked further enhancement of Akt activation [69].

Subsequent studies, utilizing similar experimental designs, confirmed and expanded these findings and directly tested the functional importance of Akt activation in the cardioprotective effect of these two PARP inhibitors (HO-3089 and 4-hydroxyquinazoline). Pharmacological inhibition of PI3-kinase by wortmannin or LY294002 reduced the PARP inhibitor-elicited increase of Akt phosphorylation during ischemia-reperfusion, and significantly diminished the PARP-inhibitor-mediated recovery of ATP and creatine phosphate in the reperfused hearts. Moreover, pharmacological inhibition of PI3-kinase also diminished the protective effect of PARP inhibitors on infarct size and the recovery of cardiac function in this model [70]. In the same study, the Akt downstream target GSK-3β was also evaluated; this effector was found to be activated by the PARP inhibitors during ischemia-reperfusion [70]. Interestingly, wortmannin or LY294002, on their own (i.e., in the absence of PARP inhibitor treatment) failed to increase infarct size or suppress cardiac function during ischemia-reperfusion, nor did they affect the basal level of Akt phosphorylation suggesting that (a) the basal and the PARP-inhibitor-induced Akt phosphorylation involves fundamentally different mechanisms; the former not being dependent on PI3-kinase, while the latter is PI3-kinase dependent and (b) the low basal level of Akt activation that occurs during ischemia-reperfusion is unable to significantly counteract cardiac reperfusion injury (as opposed to the PARP-inhibitor-induced activation of Akt, which is clearly cardioprotective) [70].

Similar findings were obtained with PARP inhibitors of various structural classes in hypoxic-reoxygenated cardiac myocytes (in vitro) and in ischemic-reperfused hearts (in vitro or in vivo) in several additional studies. For example, Palfi and colleagues evaluated the effect of the PARP inhibitor L-2286 in an ischemia-reperfusion model in Langendorff perfused rat hearts and in an isoproterenol-induced myocardial infarction model in rats. In addition to cardiac energy metabolism, oxidative damage and (in the in vivo experiment, infarct size), and the phosphorylation state of Akt and MAPK cascades were also monitored. As expected, L-2286 exerted a significant protective effect against ischemia-reperfusion-induced myocardial injury in both experimental models. In addition, in line with prior observations made with other PARP inhibitors, L-2286 enhanced the ischemia-reperfusion-induced activation of Akt. Moreover, it also induced an activation of extracellular signal-regulated kinase and p38-MAPK, while c-Jun N-terminal kinase activation was repressed [71]. Similarly, the PARP inhibitor DPQ’s protective effect in another rat isolated heart model against ischemia-reperfusion was found to be associated with a marked increase in Akt phosphorylation [72]. Moreover, in H9c2 cardiomyocytes in vitro, the PARP inhibitor 5-AIQ enhanced the oxidative stress-induced activation (phosphorylation) of Akt (as well as of GSK-3β) and these actions conferred a cytoprotective effect; inhibiting the Akt/GSK-3β pathway by LY294002 significantly attenuated the cytoprotective effect of 5-AIQ [73] and similar effects were subsequently reported with the PARP inhibitor KR-33889 in the same cardiomyocyte cell line in vitro; the data presented in the paper show that presence of the PARP inhibitor extended the duration of Akt activation after hydrogen peroxide exposure [74].

The cardioprotective effect of PARP inhibitors goes beyond protection in ischemic or hypoxic cardiomyocytes or hearts, and includes storage-mediated injury, as well as various forms of chronic heart failure (including septic, diabetic, and cardiotoxic drug-induced) [75,76,77,78,79,80,81,82,83,84,85]. Several studies have evaluated whether this protective action in these models is also associated with (and/whether it is dependent on) Akt activation. In a model of cardiac dysfunction induced by cold storage (6 h of storage in Celsior solution), the presence of the PARP inhibitor INO-1153 in the storage fluid afforded a better subsequent functional recovery, and this effect was associated with an increased Akt activation in these hearts [86]. The cardioprotective effect of the PARP inhibitor was abrogated by wortmannin co-treatment [86]. Moreover, in a model of doxorubicin-induced heart failure, the protective effect of L-2286 was associated with an activation of Akt [87], and similar effects were noted with a PARP inhibitor of a different structural class (BPG-15) in a model of imatinib-induced cardiomyopathy [88]. In addition, in a model of “hyperglycemia” in vitro (elevated extracellular glucose exposure of H9c2 cells), siRNA-mediated PARP1 depletion was found to increase the levels of phosphorylated Akt (PARP1 depletion also provided significant cytoprotective effects) [89]. The beneficial cardiac effects of L-2286 discussed earlier were also found to extend to the post-infarction cardiac remodeling stage (8 weeks of follow-up was performed); however, at these later time points, the Akt activating effect was no longer detectable (in contrast, the protective effects were attributable to effects on various protein kinase C isoforms) [90]. However, in a model of hypertension-induced cardiac remodeling in the rat, the beneficial effects of the PARP inhibitor L-2286 against cardiac remodeling and vascular dysfunction could be attributed, at least in part, to Akt activation (and in part to effects of MAP kinase pathways) [91,92].

It should be stressed that the phenomenon of Akt-activation-mediated cardioprotection is not at all unique to PARP inhibitors. The fact that Akt activation, in general, activates cytoprotective pathways is a well-known phenomenon (see prior sections); and specifically, the cardioprotective (or, more generally, organ-protective as it also extends to various organs including the brain, kidney, and others) effect of a wide set of therapeutic interventions, including ischemic pre- and post-conditioning, calcium antagonists, beta blockers, caffeine, erythropoietin, insulin, vitamin E, and many others has been attributed, partially or wholly, to a set of cellular effects (including protection against mitochondrial dysfunction, reduction of cellular reactive oxygen species generation, induction of cytoprotective, and antiapoptotic pathways, etc.) that are, wholly or partly, the consequences of Akt activation [93,94,95,96].

The phenomenon of PARP-inhibitor mediated Akt activation (and associated cytoprotective and organ-protective effects) has also been demonstrated in various non-cardiac experimental systems exposed to ischemia or hypoxia. For instance, the first-generation PARP inhibitor 3-aminobenzamide (3-AB) was found to exert beneficial effects that are (at least in part) dependent on Akt activation in a rodent model of ischemic stroke. In Sprague-Dawley rats subjected to middle cerebral artery occlusion, 3-AB increased and prolonged Akt activation in the brain, and this effect was associated with smaller infarct volume, lower brain water content, less TUNEL staining, less cleavage of caspases, and lower induction of various adhesion molecules [97]. Similarly, in an in vitro study, nicotinamide—which is a PARP inhibitor of low potency, but it also has a host of additional pharmacological actions [98]—exerted cytoprotective effects in an anoxia model in cultured rat hippocampal neurons; its beneficial effects were associated with Akt activation, and were attenuated by inhibition of Akt phosphorylation using pretreatment with wortmannin or LY294002 [99]. Application of wortmannin (500 nM) or LY294002 (10 μM) without anoxia was not toxic to the neurons, suggesting that basal Akt activity does not regulate cell viability or survival (in contrast to the PARP-inhibitor-inducible increase in Akt activity, which does) [99].

Acute urinary retention is a condition with a significant hypoxia-associated component. In a rat model of acute urinary retention induced by bladder distension, bladder apoptosis was reduced by inhibition of PARP activation with 3-aminobenzamide treatment. The PARP inhibitor also increased the levels of ATP and NAD^+^, phosphorylation of Akt, and Bcl-2/Bax ratio, and significantly reduced the activation of caspase 3 [100]. Since in this study it was not evaluated whether pharmacological inhibition of Akt activation counteracts the effects of 3-aminobenzamide, it is not possible to determine to what extent Akt activation contributed to the observed bladder-protective effects.

The cytoprotective effect of PARP inhibition (using 4-hydroxy-quinazoline) was also associated with Akt activation in an acute rat kidney rejection model [101]. Moreover, the cytoprotective effect of a clinically approved (for oncological indications) PARP inhibitor (olaparib) was also associated with Akt activation in hypoxia-reoxygenation induced acute retinal injury model [102]. However, similarly to the acute urinary retention experiments discussed above, the functional importance of Akt activation was not directly tested in these studies. In the latter model, olaparib treatment induced alterations in a host of additional pathways (MAPKs, HIF1α, Nrf2, and NFκB) [102]; thus, it is conceivable that these effects may have also contributed to its beneficial effects.

### 2.5. Mechanism of PARP Inhibition Induced Akt Activation

All aforementioned in vitro, ex vivo, and in vivo results indicate that inhibition of nuclear protein PARylation reaction catalyzed by PARP1 somehow triggers the activation of cytoplasmically localized Akt. Recently, we provided in vitro experimental evidences for a mechanism (Figure 2), which may account for the Akt activating effect of PARP1 inhibition under oxidative stress conditions, and is consistent with the experimental and clinical data on PARP inhibitors [12]. According to it, PARP1 activated by DNA strand breaks PARylates itself, thereby creating a scaffold that recruits ataxia telangiectasia mutated kinase (ATM) and NEMO [45]. Binding to PAR strands activates ATM that phosphorylates itself and NEMO [46]. If excess PARP activation is prevented by a pharmacological inhibitor or genetic manipulation, ATM is not PARylated [103], translocates to the cytoplasm in the form of a complex with NEMO, and at least in part associates with the mitochondrial outer membrane. There, the complex recruits Akt and mTOR, and together with them forms a signalosome that activates Akt, leading to activation of downstream pro-survival pathways (Figure 2) [12].

Several elements of this model have been described previously. Independent studies reported that ATM mediates Akt activation in cancer cell lines, in insulin treated myocytes, and under cellular stress conditions [104,105,106]. Defective ATM linked activation of Akt was found to be involved in diabetes and neurodegenerative diseases [106]. Furthermore, direct PARylation of ATM by PARP1 was indicated in independent reports [103,107]. PARP activity dependent formation and nuclear-to-cytosolic translocation of ATM-NEMO complex was suggested as the molecular mechanism underlying genotoxicity induced NFκB regulated apoptosis [45,108]. As for the model’s remaining elements, we demonstrated an at least partial mitochondrial localization for the proposed ATM-NEMO-Akt-mTOR signalosome, and proved that PARP1 inhibitor’s cytoprotective effect is abolished by silencing any components of the signalosome [12]. As a partial independent confirmation of latter results, PARP1 inhibition was shown to induce synthetic lethality in ATM deficient cells [109]. However, by showing that continuously active Akt rescued these cells [12], we demonstrated that the said synthetic lethality is predominantly resulted from insufficient Akt activation in ATM suppressed cells. These data emphasize the crucial importance of Akt activation in PARP1 inhibition’s cytoprotective property that can lead to undermining PARP1 inhibition’s cytostatic effect on breast cancer gene (BRCA) mutations carrying cancers [110]. Independent reports support this notion by demonstrating the increase of PARP1 inhibitors’ cytotoxicity by Akt inhibitors [8,11,111].

## 3. PARP-Akt Interactions in Cancer Biology

Based on PARP1’s role in DNA repair, early prognoses for the role of its inhibitors in cancer therapy were to enhance antitumor activity of radio- or chemotherapy [112]. Although these prognoses have proved to be correct for combination therapy, present use of PARP inhibitors is mostly based on their synthetic lethality with homologous recombination repair deficiency of DNA double strand breaks [113,114]. PARP1 binds to DNA strand breaks, and is trapped on the DNA when inactivated by the inhibitor causing replication fork collapse [115]. The resulting DNA double strand breaks are repaired by the error prone non-homologous end joining repair system when homologous recombination is defective due to mutation in its key elements, such as BRCA1 and BRCA2 [113,114]. Both mechanisms are deleterious for homologous recombination deficient cancer cells. Accordingly, in 2014, the United States Food and Drug Administration (FDA) approved the first PARP inhibitor, olaparib monotherapy for treating germline BRCA1/2 mutated, advanced stage ovarian cancer [116]. Since that time, three additional PARP inhibitors were approved and the therapy was extended to other types of BRCA deficient cancers (Table 1). Clinical trials expanded the use of PARP inhibitor monotherapy to cancers with defects other than BRCA mutation in the double-strand repair pathway and other forms of genomic instability, since these cancers seem to be more dependent on PARP1 to maintain genomic integrity [117].

Combination therapy uses PARP inhibitors together with radiotherapy, immune therapy, a cytotoxic agent, an angiogenesis blocker, a signaling pathway inhibitor, or a blocker of the DNA damage-response pathway [118]. In these therapeutic strategies, the PARP inhibitor sensitizes the cancer cells for the cytostatic effect of the agent(s) used in combination with it by limiting DNA damage repair. Alternatively, a targeted treatment, such as a signaling pathway inhibitor, could relieve the resistance of tumor cells toward the PARP inhibitor [119]. Up to now, no combination therapy with PARP inhibition was approved by FDA for the clinical practice, although a number of ongoing clinical trials utilize this strategy for various types of cancers [118].

Solid tumors grow in hypoxic condition and undergo metabolic reprograming to provide energy and nutrients for proliferation and survival. Accordingly, increased anaerobic glycolysis, modified oxidative phosphorylation, and mitochondrial remodeling are hallmarks of cancer cells’ phenotype [120]. This metabolic reprograming is often accompanied by an elevated production of reactive oxygen and nitrogen species (RONS) that can directly or indirectly activate Akt [121]. Additionally, various genetic events lead to activation of this pro-survival pathway. Activating mutations in PI3K, mTOR, and the Akt isoforms, and loss-of-function mutations and deletions in PTEN occur in about 27%, 8%, 3%, and 19% of all cancer cases, respectively [122]. PIK3CA encodes the p110α catalytic subunit of PI3K, and is the most frequently altered oncogene in human cancers, including endometrial, ovarian, colorectal, and breast cancers [123]. Mutations of mTOR and Akt are much less frequent, although, they were demonstrated in melanoma, renal carcinoma, bladder tumor, lung cancer, breast cancers, head and neck squamous cell carcinomas, and endometrial cancer [124]. PTEN loss represent the second most mutated tumor suppressor gene frequently occurring in various human cancers including colorectal cancer, breast cancer, glioblastoma, endometrial cancer, malignant melanoma, and prostate cancer [125]. Although their mutation frequency is quite low, gene amplification and overexpression of Akt isoforms occur more frequently. Akt1 gene amplification was found in breast, colon, pancreatic, gastric, esophageal, ovarian, and thyroid cancers, and glioblastoma while amplification of Akt2 occurred in pancreatic, ovarian, and breast cancers [126,127,128]. Akt2 overexpression was present in 40% of hepatocellular carcinoma and 57% in colorectal cancers, while Akt3 overexpression was found in prostate and breast cancers [126,127,128]. Furthermore, Akt was hyperactivated in many human cancers due to activating mutations of upstream elements such as PI3K and RAS, or loss of function mutations of p27, PTEN [129,130], and inositol polyphosphate-4-phosphatase2 [131]. An additional activating mechanism, N^6^-methyladenosine mRNA methylation of PTEN, leads to Akt activation in many cancers [132]. Certainly, Akt activation can also be triggered by upstream signaling elements, such as growth factors or cross-talking signaling pathways [130], or via the retrograde activating mTOR-PI3K feedback loop [133]. However, PI3K/Akt pathway’s oxidative regulation via glutathion, peroxiredoxins, glutaredoxins, and oxidative inactivation of PTEN remains a significant regulatory factor of Akt mediated malignant transformation [134,135]. Although inhibitors of the pathway are attractive agents in the cancer therapy, so far no Akt inhibitor has been approved by FDA. On the other hand, FDA approved four PI3K and three mTOR inhibitors (Table 1) for clinical cancer therapy, and two Akt inhibitors are in phase III clinical trials for combination therapy of breast and colorectal cancers [124].

One of the most important limitations of PARP inhibition based cytostatic therapy is the resistance of cancer cells against either PARP inhibition monotherapy or combination therapy. Various mechanisms can account for this resistance, including down-regulation of PARP1 expression, reduced cellular availability of the inhibitor, reactivation of the homologous recombination DNA repair pathway, exploiting altered cell cycle regulation, and changing the miRNA environment [136]. Additionally, activation of the PI3K-Akt pathway by PARP inhibition, a mechanism originally suggested for reducing side effects of platinum compounds and alkylating agents [10], represent a significant limitation of PARP inhibitor utilizing chemotherapy by protecting mitochondrial integrity and function, and mitigating apoptosis in the cancer cells (Figure 2). To support this notion, activation of Akt increased paclitaxel resistance in vitro [11], and negative synergism was found between the PARP inhibitor PJ-34 and cisplatin or temozolomide during a short-term combination treatment of B16F10 metastasizing melanoma cell line [137]. Additionally, in clinical trials, inhibitors of the P13K/AKT pathway potentiated cytostatic effect of PARP inhibitors in a combination therapy, indicating the clinical relevance of the concept [8,111].

## 4. Open Questions and Future Directions

At present, using PARP inhibitors in homologous recombination deficient malignancies is a well-established practice. Judging from the increasing number of clinical trials, extension of the strategy of synthetic lethality to more types of cancers can be expected. There are a number of ongoing clinical trials aiming at improving the efficacy of PARP inhibitors by combining them with platinum, taxane or alkylating chemotherapy, radiotherapy, and inhibitors of signaling pathways [138]. They intend to exploit synergistic effects of the agents or to overcome cytostatic resistance of the cancer cells. In the latter respect, ataxia telangiectasia and Rad3-related protein, mTOR, and NFκB pathway inhibitors were found to be effective [119,139,140].

Another line of improving therapeutic efficacy of PARP inhibition addresses removal of PAR polymers. Importance of the process is highlighted by the finding that complete PARG deficiency results in embryonic lethality [141]. Furthermore, PARG inhibition together with homologous recombination deficiency results in synthetic lethality [142], the same way as PARP1 inhibition does. Mutation of TARG1, the other enzyme that together with PARG removes PAR polymers from the target proteins, was indicated in a neurodegenerative and seizure disorder, and its deficiency sensitizes the cells to DNA alkylating agents [143]. These data suggest that removal of PAR polymers could be essential in DNA double strand break repair, and the PARG and TARG enzymes may represent targets for cancer therapy. More research and specific inhibitors are needed to elucidate the particulars concerning these enzymes.

Recent clinical trials aim to expand therapeutic application of PARP inhibitors to cancers of intact homologous recombination repair. To this end, the PARP inhibitors are proposed to be applied in combination with immune checkpoint blockade. The cancer cells are frequently resistant against immune checkpoint blockade that could be alleviated by the PARP inhibitor via promoting cross-presentation and modifying immune microenvironment. The PARP inhibitors are expected to increase T cell’s tumor-killing efficacy, and to activate the cancer-immunity cycle thereby increasing the sensitivity of tumor cells toward immune checkpoint blockade [144]. However, considering the diversity of the involved mechanisms, extensive research is needed to elucidate the possible interference by various networks including the PI3K-Akt cytoprotective signaling pathway.

With the advancement of whole exome sequencing, screening of cancer-related genetic aberrations is possible at a reasonable price. The technique provides genome-wide, and high-throughput results that can be used to identify synthetic lethal pairs involving epigenetic-related synthetic lethal genes. For the development of novel rational combination therapies, enhanced genetic-interaction screens are needed that assumes integration of functional interaction data with orthogonal methods [145]. Additionally, heterogeneity of the patient population limits therapy development, and questions optimality of the applied treatment, a problem which can be amended by determining microsatellite instability and tumor mutation burden [146]. All these techniques, as they become affordable, pave the way toward the realm of personalized medicine [147].

## 5. Conclusions

Accumulating clinical evidence indicates that PARP inhibitors can be successfully applied in cancers not responding to conventional treatments. To date, therapeutic application of the FDA approved PARP inhibitors utilize homologous recombination repair deficiency in these tumors. At the same time, ongoing clinical trials combining PARP inhibitors with agents targeting non-homologous recombination DNA repair systems, signaling pathways, angiogenesis, or immune checkpoint mechanisms intend to extend the therapeutic potential of PARP inhibitors well beyond their present one. However, because of complexity and redundancy of the mechanisms that regulate PARP activity, we still fail to comprehend fully the processes leading to de novo and acquired resistance toward PARP inhibitor therapy. As in vitro and in vivo experimental evidences indicate, the cytoprotective PI3K-Akt pathway is activated by PARP1 inhibition, and via multiple mechanisms, its activation induce apoptosis resistance and mitochondrial protection that limit cytostatic efficacy of PARP inhibitor mono- or combination therapy. Because of the importance of the PI3K-Akt pathway in all branches of medicine associated with oxidative stress including shock, inflammation, cardiac, and cerebral reperfusion injury besides cancer, extensive research is going on to fully comprehend its mechanisms and its interplay with other signaling pathways including those regulated by PARP inhibition. The availability of FDA approved PI3K-Akt pathway inhibitors should certainly facilitate these research efforts. Additionally, recent advancements in the field of next generation sequencing and bioinformatics could contribute to improving treatment strategies and outcomes of PARP inhibitor therapy.

## Figures and Tables

**Figure 1 cancers-12-00532-f001:**
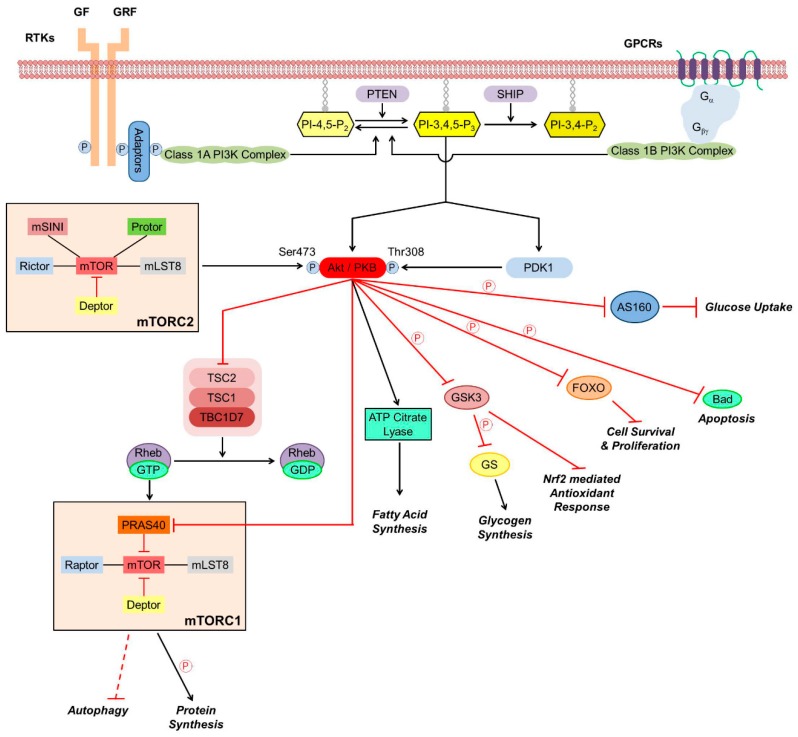
Schematic representation of the Akt signaling pathway. Pointed arrows denote activation while arrows with flat end represent inhibition. P indicates phosphorylation. AS160, Akt substrate of 160 kDa; Bad, Bcl-2 associated agonist of cell death; Deptor, dishevelled, Egl-10, and Pleckstrin domain-containing mTOR-interacting protein; FOXO, class O of forkhead box transcription factors; G_αβγ_, G protein subunits; GF, growth factor; GRF, growth-related factor; GPCR, G protein-coupled receptor; GS, glycogen synthase; GSK, GS kinase; mLST8, mammalian lethal with Sec13 protein 8; mSINI, mammalian stress-activated protein kinase interacting protein; mTOR, mammalian target of rapamycin; mTORC, mTOR complex; Nrf2, nuclear factor erythroid 2-related factor 2; PDK, 3-phosphoinositide-dependent protein kinase; PI3K, phosphatidylinositol-3 kinase; PIP_2_ phosphatidylinositol-bisphosphate; PIP_3_, phosphatidylinositol-3,4,5-trisphosphate; PKB, protein kinase B; PRAS40, proline-rich Akt substrate of 40 kDa; Protor, protein observed with rictor; PTEN, phosphatase and tensin homolog; Raptor, regulatory protein associated with mTOR; Rheb, Ras homolog enriched in brain; Rictor, rapamycin-insensitive companion of mTOR; RTK, Tyrosine kinase receptor; SHIP, phosphoinositide phosphatase; TBC1D7, TBC1 domain family member 7; TSC, tuberous sclerosis protein.

**Figure 2 cancers-12-00532-f002:**
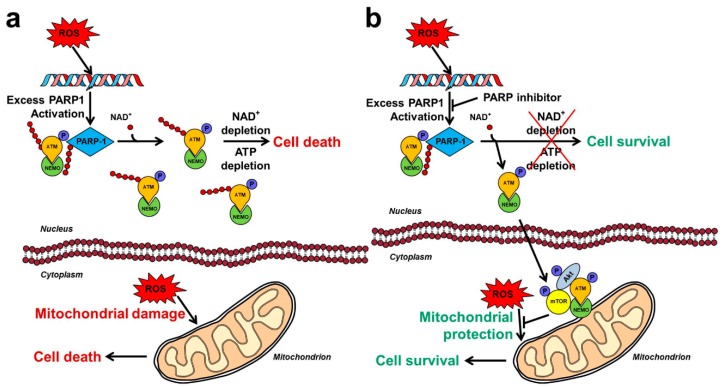
Schematic mechanisms of the PARP1/Akt interplay in oxidative stress. (**a**) Effects of oxidative stress in the absence of PARP inhibition. Oxidative stress causes DNA strand breaks that activate nuclear PARP. Excess PARP activity exhausts the substrate NAD^+^ that impairs ATP production. Attempted restoration of NAD+ levels leads to ATP depletion and eventually cell death. ATM-NEMO complexes, even if formed, are retained in the nucleus, therefore, Akt activated cytoprotective mechanisms are inactive. The oxidative stress is free to damage the mitochondria resulting in the release of pro-apoptotic intermembrane proteins. (**b**) Effects of oxidative stress in the presence of PARP inhibition. The PARP inhibitor blocks excess PARP activity, thereby prevents NAD^+^ and consequent ATP depletion. Activated ATM-NEMO complex is able to translocate to the cytoplasm and associate with the mitochondrial outer membrane. Akt and mTOR are recruited to form the ATM-NEMO-Akt-mTOR cytoprotective signalosome that prevents oxidative stress induced mitochondrial damage and activates various cell survival pathways. Pointed arrows denote activation while arrows with flat end represent inhibition. P indicates phosphorylation.

**Table 1 cancers-12-00532-t001:** Food and drug administration (FDA) approved inhibitors of enzymes relevant for the PARP-Akt interplay.

Target	Name	Company	Cancer Type
mTOR	Everolimus	Novartis	Neuroendocrine tumor, breast cancer, gastrointestinal and lung neuroendocrine tumor
mTOR	Sirolimus	Pfizer	Lymphangioleiomyomatosis
mTOR	Temsirolimus	Wyeth	Renal cell carcinoma
PI3Kδ	Idelalisib	Gilead Sciences	Chronic lymphocytic leukemia, follicular B-cell non-Hodgkin lymphoma, small lymphocytic lymphoma
PI3Kδγ	Duvelisib	Intellikine	Chronic lymphocytic leukemia/small lymphocytic lymphoma
PI3Kα	Alpelisib	Novartis	HR-positive and HER2/neu-negative breast cancer
PI3Kαδ	Copanlisib	Bayer	Relapsed follicular lymphoma
PARP1/2	Oliparib	AstraZeneca	BRCA1/2 mutated ovarian, breast, and prostate cancer
PARP1/2	Rucaparib	Clovis	BRCA1/2 mutated ovarian cancer
PARP1/2	Niraparib	Tesaro	Recurrent epithelial ovarian, fallopian tube, or primary peritoneal cancer
PARP1/2	Talazoparib	Pfizer	BRCA1/2 mutated metastatic breast cancer

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
