# Peer review of "Role of Akt Activation in PARP Inhibitor Resistance in Cancer"

_cancers, 2020, doi:10.3390/cancers12030532_

Round 1

Reviewer 1 Report

This is an interesting Review on the potential interplay between the AKT signaling pathway and PARP. The English is correct, the text is easily readable. Major problem is that the List of references is not numbered although  in the text one can find the numbers.. which makes it difficult to control appropriateness of them...

The reviewer feels that not AKT is in the focus of this review rather the PARP and PARP inhibitor resistance. Accordingly on would expect to start with PARP section, than following with AKT. Concerning AKT signaling pathway it is somehow funny that the AKT signaling pathway starts with PTEN/PI3K issue. It would be important to analyse the role of PI3K signaling in various cytokine/growth factor receptor signalings. That would be important since the resistance emerging upon PARP inhibitors may depend on GRF expression/mutation/amplification of the treated tumor type(s) which actually use that signaling pathway. It is another issue which is not mentioned here that what is the molecular epidemiology of AKT signaling pathway, what is the mutation/amplification spectrum of AKT... These informations could be also helpful to define molecular situations in a given tumor where PARP resistance could be expected. It would also be nice to see some shematic figures on AKT signaling pathway not only a dry text description. 

line 82 contains a sentence of nonsense: "oncogenic factors" what is it?

Author Response

We thank the review his/her thorough work. We think that his/her contribution has improved the manuscript considerably.

We really regret the absence of reference numbering. We have used the WORD template downloaded from the Journal’s site to prepare the manuscript, and in it the numbering was present. Unfortunately, the numbering has got lost when we uploaded the file. Nevertheless, we inserted reference numbering in the revised manuscript.   

The reviewer rightly observed that we intended to focus on the role of PARP inhibition induced Akt activation in limiting efficacy of PARP inhibitor based cancer therapy. To clarify this intention, we have changed the title to “Role of Akt activation in PARP inhibitor resistance in cancer”, and modified the Abstract and Introduction sections accordingly.

We fully agree with the reviewer that PARP inhibitor resistance of a cancer often results from expression/mutation/amplification of growth-related factors, which utilize the Akt signaling pathway and/or mutation/amplification of Akt pathway elements. However, these aspects of PARP inhibitor resistance have already been reviewed extensively. Accordingly, we started detailed description of the Akt pathway from PI3K, and mentioned the upstream processes only briefly because of space limitation.

We also agree with the reviewer that a schematic figure about Akt signalling pathway would benefit the review, and provided one in the revised manuscript.

Although space limitations compelled us to focus on how PARP inhibition induced Akt activation lead to apoptosis resistance, a process somehow neglected in reviews about PARP inhibitor based cancer therapy, we briefly referred to other aspects of PARP inhibitor resistance in the revised manuscript as requested by the reviewer.

We also corrected the sentence in line 82 to “Via phosphorylating its downstream targets, Akt can regulate metabolism, apoptosis, transcription factors, angiogenesis, and cell cycle progression.”.

Reviewer 2 Report

The authors reviewed the biological role of the PI3K-Akt pathway and the significance of the PARP-Akt interplay in shock, inflammation, cardiac and cerebral reperfusion, and cancer. They also discussed a recently discovered molecular mechanism that explains how PARP inhibition induces Akt activation and may account for apoptosis resistance and mitochondrial protection in oxidative stress and cancer.

The scope of this review manuscript is too diffusive, touching many aspects without a concise focus.

Author Response

We thank the reviewer his/her work on improving the manuscript.

We regret that he/she found it too diffusive and without a concise focus. Since better focusing was requested by reviewer #1 and the editor too, we changed the title to "Role of Akt activation in PARP inhibitor resistance in cancer", and added sentences to the Abstract and Introduction sections to clarify the aim of this review. Also, we revised the main text to focus better on how PARP inhibition induced Akt activation lead to apoptosis resistance, a process so far neglected in reviews about PARP inhibitor based cancer therapy.

Reviewer 3 Report

The review titled 'PARP inhibition and Akt signaling in cancer' by Ferenc Gallyas Jr. et al., is a well written and topical review which details the mechanisms for probable PARP resistance in cancer therapy. There are some minor grammatical/typos which need to be changed.

a) check grammar in lines 116, 158 and 252.

Author Response

We thank the reviewer his/her favourable opinion about our manuscript. We corrected the mistakes in lines 116, 158 and 252. The corrected lines are:

"transcription factors are retained in the cytoplasm in form of a complex with the 14-3-3 protein and"

"Because of PARP1’s high sensitivity toward DNA breaks and its copiousness, genotoxic stress"

"counteract cardiac reperfusion injury (as opposed to the PARP-inhibitor-induced activation of"

Round 2

Reviewer 1 Report

This is an improved version of the review. Unfortunately the issue of genetic alterations of AKT signaling pathway in cancers is still very superficially described. PTEN loss is frequent genetic alterations in cancer but activating mutations of PI3K, AKT or mTOR are also significant in various cancer types, not only those mentioned in lane427 since colorectal cancer is missing. TCGA can give you an idea on actual frequences.

This has major impact on the idea of PARP induced AKT interplay. If AKT signaling is constitutively active due to gene defects (PTEN loss or activating mutations of PI3K, AKT etc) there is a question how those physiological collaborations look  like. A safe description would be that if in a given cancer where PARP dependency is present and the AKT pathway is wild-type (no PTEN loss or activating mutations of PI3K, AKT etc) the suggested combination therapy might work. 

Author Response

We thank the reviewer his/her contribution. We think that it improved the manuscript considerably.

Originally, we did not intend to elaborate on the contribution of genetic alterations and activating mutations of the PI3K-Akt-mTOR pathway members to PARP inhibitor resistance of cancers since we intended to focus on the effect of PARP inhibitor induced activation of the pathway. However, we fully agree with the reviewer that contribution of the former is more relevant than the latter. Therefore, we tried to address the issue of genetic alterations of AKT signaling pathway as requested by the reviewer within the space limitations. We added new text and references to the revised manuscript, accordingly.

We agree with the reviewer that combination of PARP and PI3K-Akt-mTOR pathway inhibitors may have potentiality in cancer therapy. On the other hand, we do not think that usage of such combination should be restricted to cancers of wild type PI3K-Akt-mTOR pathway. In fact, PTEN deficiency was suggested to predict sensitivity toward PARP inhibitors in endometrial adenocarcinomas (Dedes et al. Sci Transl Med 2 (2010) 53r-75r). Also, in preclinical studies, combined use of PI3K inhibitor BKM120 and PARP inhibitor olaparib was successfully used in ovarian, breast and prostate cancers. However, among the PTEN deficient ovarian cancer lines, IGROV1 demonstrated significant sensitivity toward the combination therapy over the individual treatments while EFO27 was insensitive to any of the treatments (Wang et al. Oncotargets 7 (2016) 13153-13166) indicating the importance of genetic alterations additional to PTEN loss in the given cancer. Further preclinical and clinical studies should determine the clinical value of the said combination therapy.  

Reviewer 2 Report

Even though the authors changed the title to "Role of Akt activation in PARP inhibitor resistance in cancer", the content of the article remains largely the same, with only superficial modification. The scope remains wide and diffusive.

Author Response

We do regret that the revisions we have made in the first round did not meet this reviewer's expectations. In this review, we intended to increase awareness of the therapeutic limitations that the PARP-Akt interplay represent. To this end, besides the oncological models, we discuss the role of the PARP-Akt interplay in non-oncological disease conditions and their experimental models since most of the mechanistic data were acquired using latter models. By changing the title, and introducing modifications to the text requested by the other reviewers, we tried to clarify this intention. Reviewer #3 has approved concept of the review from the beginning, and reviewer #1 requested specific modifications including discussing expressional changes of and mutations to the PI3K-Akt-mTOR pathway underlying tumor resistance against PARP inhibitor therapy. Although in the latter request we did not fully meet expectations of the reviewer, he/she indicated that we are proceeding in the right direction. We hope that the present revision addresses his/her point adequately. We would have made more fundamental changes to the revised manuscript had we received more specific concerns from this reviewer. In contrast to the reviewer, we think that role of Akt activation in PARP inhibitor resistance represents a quite specific scope similar e.g. to the recent one by Revathidevi and Munirajan, Akt in cancer: Mediator and more (SeminCancerBiol 59 (2019) 80-91, and inclusion of the non-oncological models is justified by the mechanistic aspects they provide.

Round 3

Reviewer 1 Report

accept as is

Reviewer 2 Report

Besides the oncological models, the authors discuss the role of the PARP-Akt interplay in non-oncological disease conditions and their experimental models since most of the mechanistic data were acquired using latter models.